# Signal Aggregate Constraints in Additive Factorial HMMs, with Application to Energy Disaggregation

**Mingjun Zhong, Nigel Goddard, Charles Sutton**
School of Informatics
University of Edinburgh
United Kingdom
{mzhong,nigel.goddard,csutton}@inf.ed.ac.uk

## Abstract

Blind source separation problems are difficult because they are inherently unidentifiable, yet the entire goal is to identify meaningful sources. We introduce a way of incorporating domain knowledge into this problem, called *signal aggregate constraints* (SACs). SACs encourage the total signal for each of the unknown sources to be close to a specified value. This is based on the observation that the total signal often varies widely across the unknown sources, and we often have a good idea of what total values to expect. We incorporate SACs into an additive factorial hidden Markov model (AFHMM) to formulate the energy disaggregation problems where only one mixture signal is assumed to be observed. A convex quadratic program for approximate inference is employed for recovering those source signals. On a real-world energy disaggregation data set, we show that the use of SACs dramatically improves the original AFHMM, and significantly improves over a recent state-of-the-art approach.

## 1  Introduction

Many learning tasks require separating a time series into a linear combination of a larger number of "source" signals. This general problem of *blind source separation* (BSS) arises in many application domains, including audio processing [17, 2], computational biology [1], and modelling electricity usage [8, 12]. This problem is difficult because it is inherently underdetermined and unidentifiable, as there are many more sources than dimensions in the original time series. The unidentifiability problem is especially serious because often the main goal of interest is for people to interpret the resulting source signals.

For example, consider the application of energy disaggregation. In this application, the goal is to help people understand what appliances in their home use the most energy; the time at which the appliance is used is of less importance. To place an electricity monitor on every appliance in a household is expensive and intrusive, so instead researchers have proposed performing BSS on the total household electricity usage [8, 22, 15]. If this is to be effective, we must deal with the issue of identifiability: it will not engender confidence to show the householder a "franken-appliance" whose electricity usage looks like a toaster from 8am to 10am, a hot water heater until 12pm, and a television until midnight.

To address this problem, we need to incorporate domain knowledge regarding what sorts of sources we are hoping to find. Recently a number of general frameworks have been proposed for incorporating prior constraints into general-purpose probabilistic models. These include posterior regularization [4], the generalized expectation criterion [14], and measurement-based learning [13]. However, all of these approaches leave open the question of what types of domain knowledge we should include. This paper considers precisely that research issue, namely, how to identify classes

of constraints for which we often have prior knowledge, which are general across a wide variety of domains, and for which we can perform efficient computation.

In this paper we observe that in many applications of BSS, the *total signal* often varies widely across the different unknown sources, and we often have a good idea of what total values to expect. We introduce *signal aggregate constraints* (SACs) that encourage the aggregate values, such as the sums, of the source signals to be close to some specified values. For example, in the energy disaggregation problem, we know in advance that a toaster might use 50 Wh in a day and will be most unlikely to use as much as 1000 Wh. We incorporate these constraints into an additive factorial hidden Markov model (AFHMM), a commonly used model for BSS [17].

SACs raise difficult inference issues, because each constraint is a function of the entire state sequence of one chain of the AFHMM, and does not decompose according to the Markov structure of the model. We instead solve a relaxed problem and transform the optimization problem into a convex quadratic program which is computationally efficient.

On real-world data from the electricity disaggregation domain (Section 7.2.2), we show that the use of SACs significantly improves performance, resulting in a 45% decrease in normalized disaggregation error compared to the original AFHMM, and a significant improvement (29%) in performance compared to a recent state-of-the-art approach to the disaggregation problem [12].

To summarize, the contributions of this paper are: (a) introducing signal aggregate constraints for blind source separation problems (Section 4), (b) a convex quadratic program for the relaxed AFHMM with SACs (Section 5), and (c) an evaluation (Section 7) of the use of SACs on a real-world problem in energy disaggregation.

## 2   Related Work

The problem of energy disaggregation, also called non-intrusive load monitoring, was introduced by [8] and has since been the subject of intense research interest. Reviews on energy disaggregation can be found in [22] and [24].

Various approaches have been proposed to improve the basic AFHMM by constraining the states of the HMMs. The additive factorial approximate maximum a posteriori (AFAMAP) algorithm in [12] introduces the constraint that at most one chain can change state at any one time point. Another approach [21] proposed non-homogeneous HMMs combining with the constraint of changing at most one chain at a time. Alternately, semi-Markov models represent duration distributions on the hidden states and are another approach to constrain the hidden states. These have been applied to the disaggregation problems by [11] and [10]. Both [12] and [16] employ other kinds of additional information to improve the AFHMM. Other approaches could also be applicable for constraining the AFHMM, e.g., the k-segment constraints introduced for HMMs [19]. Some work in probabilistic databases has considered aggregate constraints [20], but that work considers only models with very simple graphical structure, namely, independent discrete variables.

## 3   Problem Setting

Suppose we have observed a time series of sensor readings, for example the energy measured in watt hours by an electricity meter, denoted by $Y = (Y_1, Y_2, \cdots, Y_T)$ where $Y_t \in R_+$. It is assumed that this signal was aggregated from some component signals, for example the energy consumption of individual appliances used by the household. Suppose there were $I$ components, and for each component, the signal is represented as $X_i = (x_{i1}, x_{i2}, \cdots, x_{iT})$ where $x_{it} \in R_+$. Therefore, the observation signal could be represented as the summation of the component signals as follows

$$Y_t = \sum_{i=1}^{I} x_{it} + \epsilon_t \qquad (1)$$

where $\epsilon_t$ is assumed Gaussian noise with zero mean and variance $\sigma_t^2$. The disaggregation problem is then to recover the unknown time series $X_i$ given only the observed data $Y$. This is essentially the BSS problem [3] where only one mixture signal was observed. As discussed earlier, there is no

unique solution for this model, due to the identifiability problem: component signals are exchangeable.

# 4 Models

Our models in this paper will assume that the component signals $X_i$ can be modelled by a hidden Markov chain, in common with much work in BSS. For simplicity, each Markov chain is assumed to have a finite set of states such that for the chain $i$, $x_{it} \approx \overline{\mu}_{it}$ for some $\overline{\mu}_{it} \in \{\mu_{i1}, \cdots, \mu_{iK_i}\}$ where $K_i$ denotes the number of the states in chain $i$. The idea of the SAC is fairly general, however, and could be easily incorporated into other models of the hidden sources.

## 4.1 The Additive Factorial HMM

Our baseline model will be the AFHMM. The AFHMM is a natural model for generation of an aggregated signal $Y$ where the component signals $X_i$ are assumed each to be a hidden Markov chain with states $Z_{it} \in \{1, 2, \cdots, K_i\}$ over time $t$. In the AFHMM, and variants such as AFAMAP, the model parameters, denoted by $\theta$, are unknown. These parameters are the $\mu_{ik}$; the initial probabilities $\pi_i = (\pi_{i1}, \cdots, \pi_{iK_i})^T$ for each chain where $\pi_{ik} = P(Z_{i1} = k)$; and the transition probabilities $p_{jk}^{(i)} = P(Z_{it} = j | Z_{i,t-1} = k)$. Those parameters can be estimated by using approximation methods such as the structured variational approximation [5].

In this paper we focus on inferring the sequence over time of hidden states $Z_{it}$ for each hidden Markov chain; $\theta$ are assumed known. We are interested in maximum a posteriori (MAP) inference, and the posterior distribution has the following form

$$P(Z|Y) \propto \prod_{i=1}^{I} P(Z_{i1}) \prod_{t=1}^{T} p(Y_t|Z_t) \prod_{t=2}^{T} \prod_{i=1}^{I} P(Z_{it}|Z_{i,t-1}) \quad (2)$$

where $p(Y_t|Z_t) = N(\sum_{i=1}^{I} \mu_{i,z_{it}}, \sigma_t^2)$ is a Gaussian distribution. An alternative way to represent the posterior distibution would use a binary vector $S_{it} = (S_{it1}, S_{it2}, \cdots, S_{itK_i})^T$ to represent the discrete variable $Z_{it}$ such that $S_{itk} = 1$ when $Z_{it} = k$ and for all $S_{itj} = 0$ when $j \neq k$. The logarithm of posterior distribution over $S$ then has the following form

$$\log P(S|Y) \propto \sum_{i=1}^{I} S_{i1}^T \log \pi_i + \sum_{t=2}^{T} \sum_{i=1}^{I} S_{it}^T \left( \log P^{(i)} \right) S_{i,t-1} - \frac{1}{2} \sum_{t=1}^{T} \frac{1}{\sigma_t^2} \left( Y_t - \sum_{i=1}^{I} S_{it}^T \mu_i \right)^2 \quad (3)$$

where $P^{(i)} = (p_{jk}^{(i)})$ is the transition probability matrix and $\mu_i = (\mu_{i1}, \mu_{i2}, \cdots, \mu_{iK_i})^T$. Exact inference is not tractable as the numbers of chains and states increase. A MAP value can be conveniently found by using the chainwise Viterbi algorithm [18], which optimizes jointly over each chain $S_{i1} \ldots S_{iT}$ in sequence, holding the other chains constant. However, the chainwise Viterbi algorithm can get stuck in local optima. Instead, in this paper we solve a convex quadratic program for a relaxed version of the MAP problem (see Section 5). However, this solution is not guaranteed optimal due to the identifiability problem. Many efforts have been made to provide tractable solutions to this problem by constraining the states of the hidden Markov chains. In the next section we introduce signal aggregate constraints, which will help to address this problem.

## 4.2 The Additive Factorial HMM with Signal Aggregate Constraints

Now we add Signal Aggregate Constraints to the AFHMM, yielding a new model AFHMM+SAC. The AFHMM+SAC assumes that the aggregate value of each component signal $i$ over the entire sequence is expected to be a certain value $\mu_{i0}$, which is known in advance. In other words, the SAC assumes $\sum_{t=1}^{T} x_{it} \approx \mu_{i0}$. The constraint values $\mu_{i0}$ ($i = 1, 2, \cdots, I$) could be obtained from expert knowledge or by experiments. For example, in the energy disaggregation domain, extensive research has been undertaken to estimate the average national consumption of different appliances [23].

Incorporating this constraint into the AFHMM, using the formulation from (3), results in the following optimization problem for MAP inference

$$
\underset{S}{\text{maximize}} \quad \log P(S|Y)
$$

$$
\text{subject to} \quad \left( \sum_{t=1}^{T} \mu_i^T S_{it} - \mu_{i0} \right)^2 \leq \delta_i, \quad i = 1, 2, \cdots, I, \tag{4}
$$

where $\mu_{i0}$ $(i = 1, 2, \cdots, I)$ are assumed known, and $\delta_i \geq 0$ is a tuning parameter which has the similar role as the ones used in ridge regression and LASSO [9]. Instead of solving this optimization problem directly, we equivalently solve the penalized objective function

$$
\underset{S}{\text{maximize}} \quad \mathcal{L}(S) = \log P(S|Y) - \sum_{i=1}^{I} \lambda_i \left( \sum_{t=1}^{T} \mu_i^T S_{it} - \mu_{i0} \right)^2, \tag{5}
$$

where $\lambda_i \geq 0$ is a complexity parameter which has a one-to-one correspondence with the tuning parameter $\delta_i$. In the Bayesian point of view, the constraint terms could be viewed as the logarithm of the prior distributions over the states $S$. Therefore, the objective can be viewed as a log posterior distribution over $S$. Now the Viterbi algorithm is not applicable directly since at any time $t$, the state $S_{it}$ depends on all the states at all time steps, because of the regularization terms which are non-Markovian inherently. Therefore, in the following section we transform the optimization problem (5) into a convex quadratic program which can be efficiently solved.

Note that the constraints in equation (4) could be generalised. Rather than making only one constraint on each chain in the time period $[0, T]$ (as described above), a series of constraints could be made. We could define $J$ constraints such that, for $j = 1, 2, \cdots, J$, the $j^{th}$ constraint for chain $i$ is: $\left( \sum_{\tau_j^{(i)} = t_{ij}^a}^{t_{ij}^b} \mu_i^T S_{i, \tau_j^{(i)}} - \mu_{i0}^j \right)^2 \leq \delta_{ij}$ where $[t_{ij}^a, t_{ij}^b]$ denotes the time period for the constraint. This could be reasonable particularly in household energy data to represent the fact that some appliances are commonly used during the daytime and are unlikely to be used between 2am and 5am. This is a straightforward extension that does not complicate the algorithms, so for presentational simplicity, we only use a single constraint per chain, as shown in (4), in the rest of this paper.

## 5 Convex Quadratic Programming for AFHMM+SAC

In this section we derive a convex quadratic program (CQP) for the relaxed problem for (5). The problem (5) is not convex even if the constraint $S_{itk} \in \{0, 1\}$ is relaxed, because $\log P(S|Y)$ is not convex. By adding an additional set of variables, we obtain a convex problem.

Similar to [12], we define a new $K_i \times K_i$ variable matrix $H^{it} = (h_{jk}^{it})$ such that $h_{jk}^{it} = 1$ when $S_{i,t-1,k} = 1$ and $S_{itj} = 1$, and otherwise $h_{jk}^{it} = 0$. In order to present a CQP problem, we define the following notation. Denote $\mathbf{1}_T$ as a column vector of size $T \times 1$ with all the elements being 1. Denote $\mu_i^* = \mathbf{1}_T \otimes \mu_i$ with size $TK_i \times 1$, where $\otimes$ is Kronecker product, then $\Lambda_i = \lambda_i \mu_i^* \mu_i^{*T}$ and $\tilde{\mu}_i = 2\lambda_i \mu_{i0} \mu_i^*$. Denote $\mathbf{e}_T$ as a $T \times 1$ vector with the first element being 1 and all the others being zero. Denote $\tilde{\pi}_i = \mathbf{e}_T \otimes \log \pi_i$ with size $TK_i \times 1$. We represent $\overrightarrow{\mu} = (\mu_1^T, \mu_2^T, \cdots, \mu_I^T)^T$ with size $\sum_i K_i \times 1$, and denote $V_t = \sigma_t^{-2} \overrightarrow{\mu} \overrightarrow{\mu}^T$ and $u_t = \sigma_t^{-2} Y_t \overrightarrow{\mu}$. We also denote $S_i = (S_{i1}^T, \cdots, S_{iT}^T)^T$ with size $TK_i \times 1$ and $S_t = (S_{1t}^T, \cdots, S_{It}^T)^T$ with size $\sum_i K_i \times 1$. Denote $H_{\cdot l}^{it}$ and $H_{l \cdot}^{it}$ as the column and row vectors of the matrix $H^{it}$, respectively.

The objective function in equation (5) can then be equivalently represented as

$$
\begin{aligned}
\mathcal{L}(S, H) &= \sum_{i=1}^{I} S_i^T \tilde{\pi}_i + \sum_{i,t,k,j} h_{jk}^{it} \log p_{jk}^{(i)} - \sum_{i=1}^{I} \left( S_i^T \Lambda_i S_i - S_i^T \tilde{\mu}_i \right) - \frac{1}{2} \sum_{t=1}^{T} \left( S_t^T V_t S_t - 2u_t^T S_t \right) + C \\
&= \sum_{i,t,k,j} h_{jk}^{it} \log p_{jk}^{(i)} - \sum_{i=1}^{I} \left( S_i^T \Lambda_i S_i - S_i^T (\tilde{\mu}_i + \tilde{\pi}_i) \right) - \frac{1}{2} \sum_{t=1}^{T} \left( S_t^T V_t S_t - 2u_t^T S_t \right) + C
\end{aligned}
$$

where $C$ is constant. Our aim is to optimize the problem

$$\underset{S,H}{\text{maximize}} \quad \mathcal{L}(S,H)$$

$$\text{subject to} \quad \sum_{k=1}^{K_i} S_{itk} = 1, S_{itk} \in \{0,1\}, i = 1, 2, \cdots, I; t = 1, 2, \cdots, T,$$

$$\sum_{l=1}^{K_i} H_{l.}^{it} = S_{i,t-1}^T, \sum_{l=1}^{K_i} H_{.l}^{it} = S_{it}, h_{jk}^{it} \in \{0,1\}. \tag{6}$$

This problem is equivalent to the problem in equation (5). It should be noted that the matrices $\Lambda_i$ and $V_t$ are positive semidefinite (PSD). Therefore, the problem is an integer quadratic program (IQP) which is hard to solve. Instead we solve the relaxed problem where $S_{itk} \in [0,1]$ and $h_{jk}^{it} \in [0,1]$. The problem is thus a CQP. To solve this problem we used CVX, a package for specifying and solving convex programs [7, 6]. Note that a relaxed problem for AFHMM could also be obtained by setting $\lambda_i = 0$, which is also a CQP. Concerning the computational complexity, the CQP for AFHMM+SAC has polynomial time in the number of time steps times the total number of states of the HMMs. In practice, our implementations of AFHMM, AFAMAP, and AFHMM+SAC scale similarly (see Section 7.2).

## 6 Relation to Posterior Regularization

In this section we show that the objective function in (5) can also be derived from the posterior regularization framework [4]. The posterior regularization framework guides the model to approach desired behavior by constraining the space of the model posteriors. The distribution defined in (3) is the model posterior distribution for the AFHMM. However, the desired distribution $\widetilde{P}$ we are interested in is defined in the constrained space $\left\{\widetilde{P} | E_{\widetilde{P}}\left(\varphi_i(S,Y)\right) \leq \delta_i\right\}$ where $\varphi_i(S,Y) = \left(\sum_{t=1}^T \mu_i^T S_{it} - \mu_{i0}\right)^2$. To ensure $\widetilde{P}$ is a valid distribution, it is required to optimize

$$\underset{\widetilde{P}}{\text{minimize}} \quad KL(\widetilde{P}(S)|P(S|Y))$$

$$\text{subject to} \quad E_{\widetilde{P}}\left(\varphi_i(S,Y)\right) \leq \delta_i, i = 1, 2, \cdots, I, \tag{7}$$

where $KL(\cdot|\cdot)$ denotes the KL-divergence. According to [4], the unique optimal solution for the desired distribution is $\widetilde{P}^*(S) = \frac{1}{Z} P(S|Y) \exp\left\{-\sum_{i=1}^I \lambda_i \varphi_i(S,Y)\right\}$. This is exactly the distribution in equation (5).

## 7 Results

In this section, the AFHMM+SAC is evaluated by applying it to the disaggregation problems of a toy data set and energy data, and comparing with AFHMM and AFAMAP performance.

### 7.1 Toy Data

In this section the AFHMM+SAC was applied to a toy data set to evaluate the robustness of the method. Two chains were generated with state values $\mu_1 = (0, 24, 280)^T$ and $\mu_2 = (0, 300, 500)^T$. The initial and transition probabilities were randomly generated. Suppose the generated chains were $x_i = x_{i1}, x_{i2}, \cdots, x_{iT}$ ($i = 1, 2$), with $T = 100$. The aggregated data were generated by the equation $Y_t = x_{1t} + x_{2t} + \epsilon_t$ where $\epsilon_t$ follows a Gaussian distribution with zero mean and variance $\sigma^2 = 0.01$. The AFHMM+SAC was applied to this data to disaggregate $Y$ into component signals. Note that we simply set $\lambda_i = 1$ for all the experiments including the energy data, though in practice these hyper-parameters could be tuned using cross validation. Denote $\hat{x}_i$ as the estimated signal for $x_i$. The disaggregation performance was evaluated by the *normalized disaggregation error* (NDE)

$$NDE = \frac{\sum_{i,t}(\hat{x}_{it} - x_{it})^2}{\sum_{i,t} x_{it}^2}. \tag{8}$$

For the energy data we are also particularly interested in recovering the total energy used by each appliance [16, 10]. Therefore, another objective of the disaggregation is to estimate the total energy consumed by each appliance over a period of time. To measure this, we employ the following *signal aggregate error* (SAE)

$$SAE = \frac{1}{I} \sum_{i=1}^{I} \frac{|\sum_{t=1}^{T} \hat{x}_{it} - \sum_{t'=1}^{T} x_{it'}|}{\sum_{t=1}^{T} Y_t}. \tag{9}$$

In order to assess how the SAC regularizer affects the results, various values for $\mu_0 = (\mu_{10}, \mu_{20})^T$ were used for the AFHMM+SAC algorithm. Figure 1 shows the NDE and SAE results. It shows that as the Euclidean distance between the input vector $\mu_0$ and the true signal aggregate vector $\left(\sum_{t=1}^{T} x_{1t}, \sum_{t=1}^{T} x_{2t}\right)$ increases, both the NDE and SAE increase. This shows how the SACs affect the performance of AFHMM+SAC.

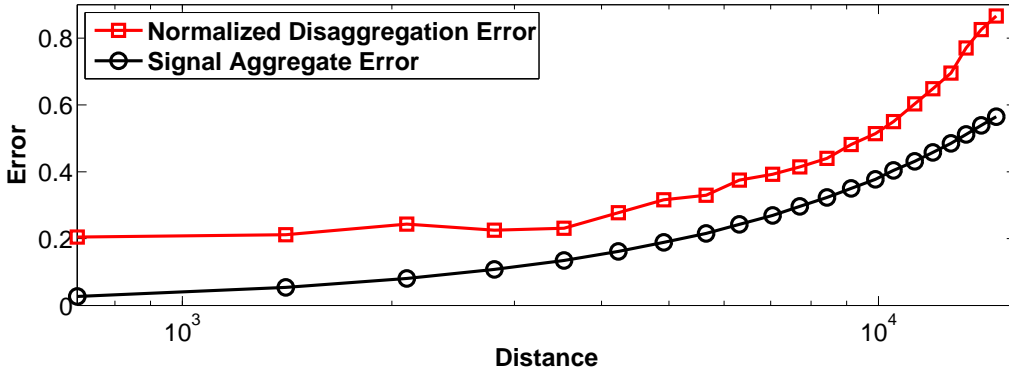

Figure 1: Normalized disaggregation error and signal aggregate error computed by AFHMM+SAC using various input vectors $\mu_{i0}$. The x-axis shows the Euclidean distance between the input vector $(\mu_{10}, \mu_{20})^T$ and the true signal aggregate vector $\left(\sum_{t=1}^{T} x_{1t}, \sum_{t=1}^{T} x_{2t}\right)^T$.

## 7.2 Energy Disaggregation

In this section, the AFHMM, AFAMAP, and AFHMM+SAC were applied to electrical energy disaggregation problems. We use the Household Electricity Survey (HES) data. HES was a recent study commissioned by the UK Department of Food and Rural Affairs, which monitored a total of 251 owner-occupied households across England from May 2010 to July 2011 [23]. The study monitored 26 households for an entire year, while the remaining 225 were monitored for one month during the year with periods selected to be representative of the different seasons. Individual appliances as well as the overall electricity consumption were monitored. The households were carefully selected to be representative of the overall population. The data were recorded every 2 or 10 minutes, depending on the household. This ultra-low frequency data presents a challenge for disaggregation techniques; typically studies rely on much higher data rates, e.g., the REDD data [12]. Both the data measured without and with a mains reading were used to compare those models. The model parameters $\theta$ defined in AFHMM, AFAMAP and AFHMM+SAC for every appliance were estimated by using 15-30 days' data for each household. We simply assume 3 states for all the appliances, though we could assume more states which requires more computational costs. The $\mu_i$ was estimated by using k-means clustering on each appliance's signals in the training data.

### 7.2.1 Energy Data without Mains Readings

In the first experiment, we generated the aggregate data by adding up the appliance signals, since no mains reading had been measured for most of the households. One-hundred households were studied, and one day's usage was used as test data for each household. The model parameters were

Table 1: Normalized disaggregation error (NDE), signal aggregate error (SAE), and computing time obtained by AFHMM, AFAMAP, and AFHMM+SAC on the energy data for 100 houses without mains. Shown are the mean±std values over days. NTC: National total consumption which was the average consumption of each appliance over the training days; TTC: True total consumption for each appliance for that day and household in the test data.

| METHODS | NDE | SAE | TIME (SECOND) |
|---|---|---|---|
| AFHMM | 0.98± 0.68 | 0.144± 0.067 | 206±114 |
| AFAMAP [12] | 0.96± 0.42 | 0.083± 0.004 | 325±177 |
| AFHMM+SAC (NTC) | 0.64± 0.37 | 0.069± 0.004 | 356±262 |
| AFHMM+SAC (TTC) | 0.36± 0.28 | 0.0015± 0.0089 | 260±108 |

estimated by using 15-26 days' data as the training data. In future work, it would be straightforward to incorporate the SAC into unsupervised disaggregation approaches [11], by using prior information such as national surveys to estimate $\mu_0$. The AFHMM, AFAMAP and AFHMM+SAC were applied to the aggregated signal to recover the component appliances. For the AFHMM+SAC, two kinds of total consumption vectors were used as the vector $\mu_0$. The first, the national total consumption (NTC), was the average consumption of each appliance over the training days across all households in the data set. The second, for comparison, was the true total consumption (TTC) for each appliance for that day and household. Obviously, TTC is the optimal value for the regularizer in AFHMM+SAC, so this gives us an oracle result which indicates the largest possible benefit from including this kind of SAC.

Table 1 shows the NDE and SAE when the three methods were applied to one day's data for 100 households. We see that AFHMM+SAC outperformed the AFHMM in terms of both NDE and SAE. The AFAMAP outperformed the AFHMM in terms of SAE, and otherwise they performed similar in terms of NDE. Unsurprisingly, the AFHMM+SAC using TTC performs the best among these methods. This shows the difference the constraints made, even though we would never be able to obtain the TTC in reality. By looking at the mean values in the Table 1, we also conclude that AFHMM+SAC using NTC had improved 33% and 16% over state-of-the-art AFAMAP in terms of NDE and SAE, respectively. This was also verified by computing the paired t-test to show that the mean NDE and SAE obtained by AFHMM+SAC and AFAMAP were different at the 5% significance level. To demonstrate the computational efficiency, the computing time is also shown in the Table 1. It indicates that AFHMM, AFAMAP and AFHMM+SAC consumed similar time for inference.

### 7.2.2 Energy Data with Mains Readings

We studied 9 houses in which the mains as well as the appliances were measured. In this experiment we applied the models directly to the measured mains signal. This scenario is more difficult than that of the previous section, because the mains power will also include the demand of some appliances which are not included in the training data, but it is also the most realistic. The summary of the 9 houses is shown in Table 2. The training data were used to estimate the model parameters. The number of appliances corresponds to the number of the HMMs in the model. The mains measured in the test days are inputted into the models to recover the consumption of those appliances. We computed the NTC by using the training data for the AFHMM+SAC. The NDE and SAE were computed for every house and each method. The results are shown in Figure 2. For each house we also computed the paired t-test for the NDE and SAE computed by AFAMAP and AFHMM+SAC(NTC), which shows that the mean errors are different at the 5% significance level. This indicates that across all the houses AFHMM+SAC has improved over AFAMAP. The overall results for all the test days are shown in Table 3, which shows that AFHMM+SAC has improved over both AFHMM and AFAMAP. In terms of computing time, however, AFHMM+SAC is similar to AFHMM and AFAMAP. It should be noted that, by looking at Tables 1 and 3, all the three methods require more time for the data with mains than those without mains. This is because the algorithms take more time to converge for realistic data. These results indicate the value of signal aggregate constraints for this problem.

Table 2: Summary of the 9 houses with mains

| HOUSE | 1 | 2 | 3 | 4 | 5 | 6 | 7 | 8 | 9 |
|---|---|---|---|---|---|---|---|---|---|
| NUMBERS OF TRAINING DAYS | 17 | 16 | 15 | 29 | 27 | 28 | 27 | 15 | 30 |
| NUMBERS OF TEST DAYS | 9 | 9 | 10 | 8 | 9 | 9 | 9 | 10 | 10 |
| NUMBERS OF APPLIANCES | 21 | 25 | 24 | 15 | 24 | 22 | 23 | 20 | 25 |

Table 3: The normalized disaggregation error (NDE), signal aggregate error (SAE), and computing time obtained by AFHMM, AFAMAP, and AFHMM+SAC using mains as the input. Shown are the mean±std values computed from all the test days of the 9 houses. NTC: National total consumption which was the average consumption of each appliance over the training days; TTC: True total consumption for each appliance for that day and household in the test data.

| METHODS | NDE | SAE | TIME (SECOND) |
|---|---|---|---|
| AFHMM | 1.36± 0.75 | 0.069± 0.039 | 1008±269 |
| AFAMAP [12] | 1.05± 0.29 | 0.043± 0.012 | 1327±453 |
| AFHMM+SAC (NTC) | 0.74± 0.34 | 0.030± 0.014 | 1101±342 |
| AFHMM+SAC (TTC) | 0.57± 0.28 | 0.001± 0.0048 | 1276±410 |

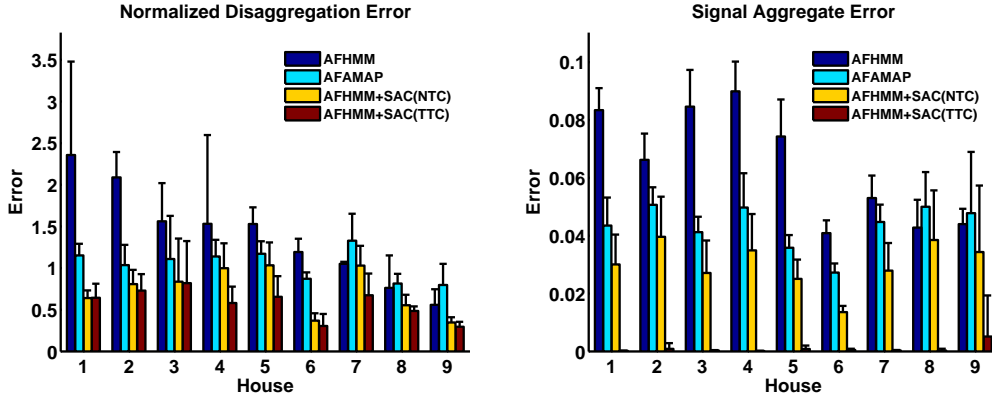

Figure 2: Mean and std plots for NDE and SAE computed by AFHMM, AFAMAP and AFHMM+SAC using mains as the input for 9 houses.

## 8 Conclusions

In this paper, we have proposed an additive factorial HMM with signal aggregate constraints. The regularizer was derived from a prior distribution over the chain states. We also showed that the objective function can be derived in the framework of posterior regularization. We focused on finding the MAP configuration for the posterior distribution with the constraints. Since dynamic programming is not directly applicable, we pose the optimization problem as a convex quadratic program and solve the relaxed problem. On simulated data, we showed that the AFHMM+SAC is robust to errors in specification of the constraint value. On real world data from the energy disaggregation problem, we showed that the AFHMM+SAC performed better both than a simple AFHMM and than previously published research.

**Acknowledgments**

This work is supported by the Engineering and Physical Sciences Research Council (grant number EP/K002732/1).

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
