[Reviews · NeurIPS 2014]

Submitted by Assigned_Reviewer_1

This paper develops a new method of performing blind source separation, by formulating the problem as an additive factorial HMM (AFHMM), and then applying signal aggregate constraints (SACs). The motivation behind this is that additional domain knowledge can be incorporated to improve the separation of the time series into components. The example used throughout the paper is energy disaggregation, where the components of domestic energy use (relating to individual appliances) can be better separated, when information relating to total (expected) usage of each appliance in a time period is incorporated. The objective function that is maximized to perform the separation (which is the log of the posterior distribution of the hidden chains given the observed data) is then transformed into a convex optimization problem. The efficacy of the method is demonstrated in practice with both a toy data set, and with real data from domestic energy output readings.

The paper is technically sound with a strong simulation section. The method, as expected, performs favorably in contrast to regular AFMM and also AFAMAP (a recently developed alternative method) - neither of which incorporate the SACs. The improvements are shown to be significant using t-tests. The author(s) also show that the speed of optimization is comparable to AFAMAP. My only question on the results section is why the author(s) used such a low noise variance in their toy example - is this to make the illustration of Figure 2 clear that they can separate the signal very well and other methods cannot? Some clarification would be good here otherwise the problem setup seems "overly" toy and rather easy. Or are these signal to noise ratios common in real data? On this note, the authors could also comment on the signal/noise ratio between the mains readings and the aggregate appliance output in the energy application.

The theoretical aspects of the paper are overall well presented and structured. The paper would perhaps benefit from a proof that the final procedure in equation (6) is indeed convex, as it is not quite clear to me using the reasoning provided. In addition, theoretical scaling arguments (with respect to data length and number of hidden states) would benefit the paper, and compliment the simulation findings. Are the methods tractable if used for much larger data sets and state spaces, and how would this contrast with AFAMAP?

The paper is well written and structured. The problem is nicely motivated in the introduction, with a clear description of relevant literature, though I am not that familiar with this problem, so cannot definitively state that nothing has been omitted here. I point to a small (potential) error on page 4 (line 202), in the unnumbered equation before (6), where I believe there should be no commas in the subscripts proceeding the object p, to be consistent with earlier notation on page 3 (unless new notation has been introduced here?)

To my knowledge the work is novel and original, though I am not an expert so cannot comment on the significance of these results to the HMM community and to the energy application studied.

Addendum: The authors satisfactorily answered my questions in the feedback phase, and I encourage them to make similar clarifications in the next version of the paper.
Summary: This paper proposed a method of performing blind source separation using Additive Factorial HMMs with Signal Aggregate Constraints. The method performs very well, when compared with the state-of-the-art, particularly with real data from an energy disaggregation problem.

Submitted by Assigned_Reviewer_18

An additive factorial hidden Markov model is developed to model electricity usage. The model incorporates the so-called signal aggegate constraint.
The quality of the paper is good, but it is really hard to read. There are too many acronyms, and some of them are really log (even 11 letters!)
Summary: It seems that the Authors are looking for a constraint to be placed on the posterior density and so they place a constraint on a prior density. They should state they have an a priori constraint and hence the posterior is constrained.

Submitted by Assigned_Reviewer_26

The paper proposes additive factorial HMM (AFHMM) with signal aggregate constraints (SAC) to deal with blind source separation problem. Authors propose convex quadratic problem for relaxed AFHMM with SAC. The technique shows promising results on both synthetic and real world data.

Pros:
- Paper is well written and has good mathematical foundation.
- Problem motivation is good
- Evaluation results indicate good boost in performance metrics, NDE and SAE

Areas of improvement:
- Results are shown as an improvement over AFHMM without SAC and AFAMAP. Both these use less data than SACAFHMM which seems unfair comparison. Are there any other approaches that could use SAC but are not as efficient in computation?
- Results in Figure 1 are not clear. It seems like for distance more than 10^4 AFAMAP beats SACAFHMM. Does this indicate that bad constraints can result in much poorer performance? Discussion seem to indicate that results are always better which seems misleading.
- I was expecting to see more complex constraints incorporated in the algorithm. For e.g. Person cannot use the heater and cooler at the same time. What about having no more than 3 out of 10 bulbs in the house active at the same time. Things like grinders having time constraints of less than 10 mins. To summarize, there could various kinds of constraints that could be specified. Is it possible to extend the algorithm to other kinds of constraints as well?
Summary: Paper is well written and provides good motivation into the problem. Future work involves extending the model further to other kinds of constraints.

Submitted by Assigned_Reviewer_40

The paper proposes an extension to the additive factorial HMM model where additional constraints are added to require that the sum of individual chain outputs to be close to some reasonable (known) aggregate value. The authors propose a convex relaxation to MAP inference and apply the algorithm to real-world data sets involves energy disaggregation, both across 100 UK homes with low frequency readings, and 10 homes with higher frequency readings (but without each appliance monitored).

The ideas presented here are fairly straightforward, and the authors use a relatively simple optimization formulation to solve these problems, but overall the work is still compelling. The authors are applying these methods to relatively large real-world data sets, and as a paper that combines both applied and algorithmic elements, I think that it is a success. The comments below could help to improve the paper in my view:

A motivation for AFAMAP to enforce the constraint that one chain stages state at a time is that this can improve the relaxation to achieve integer solutions more often. Does the SACAFHMM optimization formulation also accomplish a similar thing, or do the resulting solutions still have lower error despite have non-integral solutions? Some discussion of these points would be very helpful as it would also further elucidate the distinctions.

Fundamentally, it seems like the AFAMAP and SACAFHMM ideas are orthogonal: one could just as easily include the SACAFHMM contraints in the optimization formulation for AFAMAP. While I don't expect feedback on this during the author response, I would strongly encourage them to try this for a final version.

The authors do seem to conflate models and optimization procedures a few times. For instance, comparing the results of "AFHMM" versus "AFAMAP" isn't being very precise: AFAMAP is an algorithm for (approximately) computing MAP solutions to the AFHMM, and I believe by AFHMM here they really mean alternating Viterbi optimization. Is this correct? This also gets to the point above, where the strategies of the different algorithms presented here could easily be combined and might in fact perform better than the current approaches.
Summary: The paper proposes a simple extension and optimization to solve a variant of the AFHMM, but the results on real world data is compelling here, and overall the paper looks reasonable.
Author Feedback
Author rebuttal: We thank all reviewers for their very helpful comments.

REVIEWER 1

The only question on the results is our use of a low noise variance in the toy data. We will provide an analysis of sensitivity of results to noise variance in the final paper, including calculation of signal/noise ratio in the real data and how that relates to the toy data results.

* Convexity: This is observed by noting that the objective function in (6) has a linear component and a quadratic component, and the constraint is linear, so this is a convex quadratic programming problem (and hence convex).

* Scaling: Both AFAMAP and SACAFHMM are convex quadratic programming (CQP) problems. CQP has known computational bounds (polynomial in the number of time steps times the total number of states of the appliances, which we will note in the final paper.) In practice our implementations of SACAFHMM and AFAMAP scale similarly, since the optimization problem for SACAFHMM has the same number of variables as AFAMAP, although AFAMAP actually has more constraints.

REVIEWER 18:

We thank the reviewer's positive comments.

REVIEWER 26

* Fairness of our comparison: We are unaware of any previous research that uses SAC. The point of our method is to provide a way to incorporate prior knowledge (i.e., the SAC) into the separation problem. Therefore, our experiments were specifically designed to compare the effect of having prior knowledge to not having it.

* Figure 1: Yes, this is correct, this experiment indicates that extremely inaccurate constraints are worse than no constraints at all. We would argue that this is an expected result. The results on the energy data indicate that in real-world scenarios, it is possible to obtain constraint values that are reliable enough to improve performance.

* More complex constraints: This is an interesting idea, but the tradeoff is how complex they make the corresponding optimization problem. This is an interesting avenue for future work.

REVIEWER 40

* Non-integral solutions: The AFAMAP does help to improve the relaxation to achieve integer solutions more often. Indeed, in our experiments, we observed that SACAFHMM also prefers integer and sparse solutions, which actually does improve the results over AFHMM and AFAMAP as shown in the paper.

* AFAMAP and SACAFHMM ideas are orthogonal: We agree. We will try this and report the results in the final paper.

* "AFAMAP is an algorithm for (approximately) computing MAP solutions to the AFHMM": In our view, the models for AFAMAP and AFHMM are also different, as we would view the one-at-a-time constraint as an additional modelling assumption within AFAMAP. Yes, the AFHMM results in our paper use alternating Viterbi.